# Can Generative AI Solve Your In-Context Learning Problem? A Martingale Perspective

Andrew Jesson*†      Nicolas Beltran-Velez‡      David Blei†‡

## Abstract

This work is about estimating when a conditional generative model (CGM) can solve an in-context learning (ICL) problem. An in-context learning (ICL) problem comprises a CGM, a dataset, and a prediction task. The CGM could be a multi-modal foundation model; the dataset, a collection of patient histories, test results, and recorded diagnoses; and the prediction task to communicate a diagnosis to a new patient. A Bayesian interpretation of ICL assumes that the CGM computes a posterior predictive distribution over an unknown Bayesian model defining a joint distribution over latent explanations and observable data. From this perspective, Bayesian model criticism is a reasonable approach to assess the suitability of a given CGM for an ICL problem. However, such approaches—like posterior predictive checks (PPCs)—often assume that we can sample from the likelihood and posterior defined by the Bayesian model, which are not explicitly given for contemporary CGMs. To address this, we show when ancestral sampling from the predictive distribution of a CGM is equivalent to sampling datasets from the posterior predictive of the assumed Bayesian model. Then we develop the generative predictive $p$-value, which enables PPCs and their cousins for contemporary CGMs. The generative predictive $p$-value can then be used in a statistical decision procedure to determine when the model is appropriate for an ICL problem. Our method only requires generating queries and responses from a CGM and evaluating its response log probability. We empirically evaluate our method on synthetic tabular, imaging, and natural language ICL tasks using large language models.

## 1 Introduction

An in-context learning (ICL) problem involves a conditional generative model (CGM), a dataset, and a prediction task [1, 2]. The CGM could be a pre-trained foundation model. The dataset might consist of patient histories, test results, and diagnoses. The prediction task could be providing a diagnosis to a new patient based on their history and test results [3]. This is a complex problem that requires both accurate diagnosis and proper communication to the patient. This complexity makes it difficult to evaluate whether the model is suitable for the dataset and prediction task.

An interpretation of ICL sees a CGM prompted with in-context examples as producing data (either responses or examples of the prediction problem) from a posterior predictive under a Bayesian model. A natural question arises when we accept this premise, "Is the Bayesian model a good model for the prediction problem?" This question is what Bayesian model criticism tries to answer. This field has produced many methods but, they typically assume access to components defined by the Bayesian model, like the likelihood and posterior. In this work we show how to do model criticism in ICL using contemporary generative AI. Specifically, we demonstrate how to implement posterior predictive checks (PPCs) [4, 5] and their cousins [6, 7] when we only have access to the predictive distribution. The result is a practical and interpretable test on whether a model can solve an ICL problem.

---

*Correspondence to adj2147@columbia.edu. † Department of Statistics, Columbia University. ‡ Department of Computer Science, Columbia University.

Safe Generative AI Workshop (NeurIPS 2024).

```
Input: proof once again that if the
       filmmakers just follow the books
Label: negative
Input: is impressive
Label: positive
Input: the top japanese animations
Label: positive
Input: a spoof comedy
Label: positive
```

(a) SST2 ICL dataset $\mathrm{x}^n$

```
Input: a) Should teens use the diet
          plans on tv?
       b) Can you help me with a diet
          plan?
Label: different
Input: a) What's a good way to address
          back pain?
       b) How can I cure my back pain?
Label: similar
```

(d) MQP ICL dataset $\mathrm{x}^n$

```
Input: follows the formula , but throws
       in too many conflicts to keep
       the story compelling .
```

(b) query z.

```
Input: a) Can dementia cause ANS dysfunction?
          If so how?
       b) How can dementia cause ANS dysfunction?
```

(e) query z.

```
Label: negative, Label: negative,
Label: negative, Label: negative,
Label: negative, Label: negative
```

(c) CGM responses y

```
Label: different, Label: different,
Label: similar, Label: similar,
Label: different, Label: similar
```

(f) CGM responses y

Figure 1: An example illustrating two ICL problems. One that the model $\boldsymbol{\theta}$ (Llama-2 7B [8]) can solve, and one that it cannot. Left: (a) examples from the SST2 task [9] comprising part of an ICL dataset $\mathrm{x}^n$, (b) a new query z, and (c) some responses y sampled from the CGM $p_{\boldsymbol{\theta}}(\mathrm{y} \mid \mathrm{z}, \mathrm{x}^n)$ when prompted with the dataset and query. The true label is "negative" and the CGM responds correctly. Right: the same format but the dataset (d) and query (e) are taken from the MQP task [10]. Here, the true label is "similar," but the model responds incorrectly with "different" half of the time.

## 2 What is an in-context learning problem?

An ICL problem is a tuple $(\mathrm{f}^*, \mathrm{x}^n, \boldsymbol{\theta})$ comprising a prediction task $\mathrm{f}^*$, a dataset $\mathrm{x}^n$, and a conditional generative model (CGM) $\boldsymbol{\theta}$. The prediction task is generalized as providing a response y to a query z. The set of valid responses to a user query implies a distribution over responses $p(\mathrm{y} \mid \mathrm{z}, \mathrm{f}^*)$. The dataset $\mathrm{x}^n = \{(\mathrm{z}_i, \mathrm{y}_i)\}_{i=1}^n$ comprises $n$ query and response examples of the prediction task; $\mathrm{z}_i, \mathrm{y}_i \sim p(\mathrm{z}, \mathrm{y} \mid \mathrm{f}^*)$. A practical abstraction decomposes queries and responses into elements called tokens. As such, queries and responses—by extension, examples and datasets—are represented as sequences of tokens. For example, $(\mathrm{z}, \mathrm{y}) \equiv (\mathrm{t}_1^z, \mathrm{t}_2^z, \dots, \mathrm{t}_1^y, \mathrm{t}_2^y, \dots) \equiv (\mathrm{t}_1^x, \mathrm{t}_2^x, \dots)$. A CGM $\boldsymbol{\theta}$ defines a predictive distribution over the next token in an example $\mathrm{t}_j^x$ given previous example tokens and $\mathrm{t}_{<j}^x$, and a tokenized dataset; $p_{\boldsymbol{\theta}}(\mathrm{t}_j^x \mid \mathrm{t}_{<j}^x, \mathrm{x}^n)$. By ancestral sampling, the CGM effectively defines additional predictive distributions over responses $p_{\boldsymbol{\theta}}(\mathrm{y} \mid \mathrm{z}, \mathrm{x}^n)$, examples $p_{\boldsymbol{\theta}}(\mathrm{z}, \mathrm{y} \mid \mathrm{x}^n)$, and datasets $p_{\boldsymbol{\theta}}(\mathrm{x} \mid \mathrm{x}^n)$. Figure 1 illustrates examples from two different ICL tasks. Figures 1a to 1c gives an example from the SST2 sentiment prediction task for which Llama-2 7B frequently yields accurate answers. Figures 1d to 1f gives an example from the medical questions pairs (MQP) prediction task for which Llama-2 7B yields random answers on average. Next, we give several reasons why model generated responses or examples may be inappropriate for the ICL task $\mathrm{f}^*$.

## 3 What is a model?

Again let $\boldsymbol{\theta}$ denote a model, but now the model could be Bayesian linear regression, a Gaussian process, or perhaps a large language model (LLM). A model defines a joint distribution $p_{\boldsymbol{\theta}}(\mathrm{x}, \mathrm{f})$ over observable data $\mathrm{x} = \{\mathrm{x}_1, \mathrm{x}_2, \dots\} = \{(\mathrm{z}_1, \mathrm{y}_1), (\mathrm{z}_2, \mathrm{y}_2), \dots\}$ and latent explanations f. The notation f denotes both tasks and explanations, but we will clearly distinguish between them. The model joint distribution factorizes as $p_{\boldsymbol{\theta}}(\mathrm{x}, \mathrm{f}) = p_{\boldsymbol{\theta}}(\mathrm{x} \mid \mathrm{f})p_{\boldsymbol{\theta}}(\mathrm{f})$, where $p_{\boldsymbol{\theta}}(\mathrm{f})$ is the prior over explanations and $p_{\boldsymbol{\theta}}(\mathrm{x} \mid \mathrm{f})$ is the likelihood of the dataset given an explanation. From a frequentist perspective, the prior distribution over f would be ignored and a model would define a set of distributions over datasets indexed by f; $\{p_{\boldsymbol{\theta}}(\mathrm{x} \mid \mathrm{f}) : \mathrm{f} \in \mathcal{F}\}$. A model $\boldsymbol{\theta}$ alongside data $\mathrm{x}^n$ further defines the posterior $p_{\boldsymbol{\theta}}(\mathrm{f} \mid \mathrm{x}^n)$ and posterior predictive $p_{\boldsymbol{\theta}}(\mathrm{x} \mid \mathrm{x}^n) = \int p_{\boldsymbol{\theta}}(\mathrm{x} \mid \mathrm{f}) \, d\mathrm{P}_{\boldsymbol{\theta}}(\mathrm{f} \mid \mathrm{x}^n)$ distributions, which specify the conditional distributions of explanations and new observations given the observed data. A deeper discussion on the component distributions defined by a model is given in Appendix A.

**Are CGMs Models?** Modern CGMs often lend access to only the marginal $p_{\boldsymbol{\theta}}(\mathrm{x})$ or posterior predictive $p_{\boldsymbol{\theta}}(\mathrm{x} \mid \mathrm{x}^n)$ rather than an explicit representation of f. Why, then, can we still discuss latent

variables like f? We justify this with two key assumptions: First, if the model $p_{\boldsymbol{\theta}}$ is exchangeable (i.e., the distribution $p_{\boldsymbol{\theta}}(\mathrm{x})$ is invariant to permutations of the data), de Finetti's theorem [11] guarantees the existence of such a latent variable f. Therefore, assuming we adopt a unique representation of f, there is no issue in writing $p_{\boldsymbol{\theta}}(\mathrm{x}, \mathrm{f})$. Alternatively, if $p_{\boldsymbol{\theta}}(\mathrm{x})$ approximates an exchangeable distribution $p(\mathrm{x})$, as is the case with ICL problems, then we can treat the statement $p_{\boldsymbol{\theta}}(\mathrm{x}, \mathrm{f})$ as a convenient abuse of notation, meant to represent $p(\mathrm{x}, \mathrm{f})$. Throughout, we assume that either of these conditions hold.

## 4   A model is a choice to be criticised

A model $\boldsymbol{\theta}$ over an observation space $\mathcal{X}$ is used to make inferences based on observations $\mathrm{x}^n$. These inferences can include probabilities of the next word in a sequence, model uncertainty, and other quantities of interest. However, since the model is a choice made by the practitioner, there is no guarantee that these inferences reflect reality or adequately model the data. For example, a randomly initialized LLM can be used to make inferences about next word probabilities, but those inferences are meaningless for modeling natural language. As models become more complex and widely used, it is crucial to understand when they can be trusted.

Much of the discussion around the reliability of CGMs has focused on "hallucination" detection, prediction, and mitigation [12–35]. An interesting subset of these methods are based on uncertainty quantification where inferences about the variability of responses from the posterior predictive distribution [36–39], or about the variability of explanations [33, 35, 40–42] are used to predict when a model may hallucinate. However, these methods do not address the fundamental question of when to trust those inferences so they are susceptible to failure if the model is not appropriate for a task.

A growing body of work is formalizing the connection between ICL with pre-trained CGMs and Bayesian inference [35, 43–48]. Notably, the works of Jesson et al. [35], Fong et al. [45], Lee et al. [46], Falck et al. [47], Ye et al. [48] show how to transform Bayesian functionals of the model likelihood $p_{\boldsymbol{\theta}}(\mathrm{x} \mid \mathrm{f})$ and model posterior $p_{\boldsymbol{\theta}}(\mathrm{f} \mid \mathrm{x}^n)$ into functionals of the model predictive distribution $p_{\boldsymbol{\theta}}(\mathrm{x} \mid \mathrm{x}^n)$, which can be computed by contemporary CGMs. These works pave the way for using Bayesian model criticism techniques such as posterior predictive checks as a response to our ressearch question. In the following we formalize how this is done.

## 5   Posterior predictive checks are model critics for ICL problems.

Posterior predictive checks (PPCs) [5–7, 49] are Bayesian model criticism methods that use the posterior predictive to evaluate a model's capability to make inferences from observations. Model capability is measured by the posterior predictive $p$-value, based on the hypothesis that the data are generated according to the model $\boldsymbol{\theta}$. Following Moran et al. [7], we assume access to main $\mathrm{x}^n$ and holdout $\mathrm{x}^{\text{test}}$ sets, both distributed according to the reference likelihood $p(\mathrm{x} \mid \mathrm{f}^*)$. A class of PPCs assess how well a model $\boldsymbol{\theta}$ fit to $\mathrm{x}^n$ explains the holdout data $\mathrm{x}^{\text{test}}$. To measure fit, a PPC defines a discrepancy function, such as the negative log marginal model likelihood $g_{\boldsymbol{\theta}}(\mathrm{x}, \mathrm{x}^n) := -\sum_{\mathrm{z}_i, \mathrm{y}_i \in \mathrm{x}} \log p_{\boldsymbol{\theta}}(\mathrm{z}_i, \mathrm{y}_i \mid \mathrm{x}^n)$, or the negative log model likelihood $g_{\boldsymbol{\theta}}(\mathrm{x}, \mathrm{f}) := -\sum_{\mathrm{z}_i, \mathrm{y}_i \in \mathrm{x}} \log p_{\boldsymbol{\theta}}(\mathrm{z}_i, \mathrm{y}_i \mid \mathrm{f})$. Higher values indicate poor model fit, while lower values suggest the model explains the data well.

Defining a goodness-of-fit measure is only half of the story. A PPC needs to define what are relatively high or relatively low values of the discrepancy function. To do this, a reference distribution of values is defined by measuring the discrepancy function over datasets sampled from the model posterior predictive distribution. The posterior predictive $p$-value is then evaluated as

$$p_{\text{ppc}} := \iint \mathbb{1}\big\{g_{\boldsymbol{\theta}}(\mathrm{x}, \cdot) \geq g_{\boldsymbol{\theta}}(\mathrm{x}^{\text{test}}, \cdot)\big\} d\mathrm{P}_{\boldsymbol{\theta}}(\mathrm{x} \mid \mathrm{f}) d\mathrm{P}_{\boldsymbol{\theta}}(\mathrm{f} \mid \mathrm{x}^n). \tag{1}$$

The PPC compares the discrepancy function value for holdout data $g_{\boldsymbol{\theta}}(\mathrm{x}^{\text{test}}, \cdot)$ with its distribution under the model $g_{\boldsymbol{\theta}}(\mathrm{X}, \cdot)$. If the model-generated data often has a greater or equal discrepancy than the holdout data, we can be confident the model explains the holdout data well. Conversely, if the holdout data's discrepancy is frequently higher, we should be less confident in the model's ability to explain it, raising doubts about the model's ability to solve the ICL problem. Algorithm 2 in Appendix C describes a $p_{\text{ppc}}$ estimator.

# 6 The generative predictive p-value and how to estimate it.

Modern CGMs—such as LLMS—do not provide an explicit representaion of the joint distribution over observations and explanations. For discrepancy functions that depend on f this is a problem.

Our solution to the inaccessibility of component distributions relies on the intuition that a large enough dataset $x^N := \{z_i, y_i\}_{i=1}^N$ generated by the likelihood $x^N \sim p_{\boldsymbol{\theta}}(x \mid f)$ contains roughly the same information as the explanation f itself for an identifiable Bayesian model. Thus, it makes sense to express functions of f, like the model likelihood $p_{\boldsymbol{\theta}}(x \mid f)$, that are not defined by a CGM, as functions of large datasets, such as the predictive distribution $p_{\boldsymbol{\theta}}(x \mid x^N)$, which are defined.

Now, where do the additional $N - n$ examples come from if we are only given the $n$ examples comprising $x^n$? The generation of sufficiently large datasets is done by generating hypothetical completions $x^{n+1:N}$ of the observed ICL dataset $x^n$ by ancestrally sampling from the model predictive distribution $p_{\boldsymbol{\theta}}(z, y \mid x^n)$ (also called predictive resampling by Fong et al. [45]):

$$z_{n+1}, y_{n+1} \sim p_{\boldsymbol{\theta}}(z, y \mid x^n), z_{n+2}, \quad y_{n+2} \sim p_{\boldsymbol{\theta}}(z, y \mid x^n, z_{n+1}, y_{n+1}), \quad \dots$$

As the generated examples are added to the conditional of the predictive distribution after each step, this process can be thought of as reasoning toward one explanation by imagining a sequence of sets of observations that are consistent with a smaller and smaller set of explanations as the sequence length increases. As a stochastic process, it is encouraged to reason toward a different explanation each time it is run to complete $x^n$ with $N - n$ imagined examples.

Building off this intuition, we define martingale and generative predictive $p$-values below. We prove that under general conditions the martingale predictive $p$-value is equal to the posterior predictive $p$-value. We then show how to estimate the generative predictive $p$-value for a given ICL problem.

## 6.1 The martingale predictive p-value

Our method is built on Doob's theorem for estimators (Theorem 2), which helps us transform statements about the random variable $h(F)$—a function of explanations F—to statements about the random variable $\mathbb{E}[h(F) \mid X_1, X_2, \dots, X_n]$, which is a function of observations $(X_1, X_2, \dots, X_n)$. Thus, we can proceed without direct access to $p_{\boldsymbol{\theta}}(z, y \mid f)$and $p_{\boldsymbol{\theta}}(f \mid x_n)$ and define a $p$-value that depends on infinite datasets $x^\infty := (x_i, y_i)_{i=1}^\infty$ rather than f

$$p_{\text{mpc}} := \iint \mathbb{1}\{g_{\boldsymbol{\theta}}(x, x^\infty) \geq g_{\boldsymbol{\theta}}(x^{\text{test}}, x^\infty)\} d\mathrm{P}_{\boldsymbol{\theta}}(x \mid x^\infty) d\mathrm{P}_{\boldsymbol{\theta}}(x^{n:\infty} \mid x^n). \tag{2}$$

Doob's Theorem is an application of martingales, so—in line with the current literature [45–47]—we call this formulation the martingale predictive $p$-value.

The main theoretical result of this paper establishes the equality of the posterior and martingale predictive $p$-values; Equations (1) and (2). We formalize this statement in the following theorem.

**Theorem 1.** *Let* $F \sim \mathrm{P}_{\boldsymbol{\theta}}$, *and* $X_1, X_2, \dots$ *i.i.d* $\sim \mathrm{P}_{\boldsymbol{\theta}}^{\mathrm{f}}$. *Assume Conditions 1 to 3 and let,*

$$\int |\log p_{\boldsymbol{\theta}}(x^m \mid f)| d\mathrm{P}_{\boldsymbol{\theta}}(f) < \infty : \quad \forall x^m \in \mathcal{X}^m.$$

*Then,* $p_{ppc} = p_{mpc}$.

*Proof.* The proof makes use of Doob's Theorem and is presented in Appendix B. $\square$

## 6.2 The generative predictive p-value

The martingale predictive $p$-value cannot be exactly computed because it is impossible to generate infinite datasets. Thus, we define the generative predictive $p$-value that clips the limits to infinity by some feasibly large number $N$ to estimate Equation (2) as

$$p_{\text{gpc}} := \iint \mathbb{1}\{g_{\boldsymbol{\theta}}(x, x^N) \geq g_{\boldsymbol{\theta}}(x^{\text{test}}, x^N)\} d\mathrm{P}_{\boldsymbol{\theta}}(x \mid x^N) d\mathrm{P}_{\boldsymbol{\theta}}(x^{n:N} \mid x^n). \tag{3}$$

Note that the generative predictive $p$-value enables us to replace any distributions that depend on latent mechanisms f or infinite datasets $x^\infty$ with ones that depend on finite sequences. The price we pay for using finite $N$ is estimation error between $p_{\text{gpc}}$ and $p_{\text{ppc}}$. We leave the formal analysis of this error to future work.

## 6.3 CGM estimators for the generative predictive p-value

---

**Algorithm 1** $\widehat{p}_{\text{gpc}}$

---

**Require:** data $\{\mathrm{x}^n, \mathrm{x}^{\text{test}}\}$, discrepancy function $g_{\boldsymbol{\theta}}(\mathrm{x}, \mathrm{x}^N)$, # replicates M, # approx. samples N
1: **for** $i \leftarrow 1$ to M **do**
2:     $\mathrm{x}_i^N \leftarrow \mathrm{x}^n$                                                                          ▷ initialize f sample data
3:     **for** $j \leftarrow n+1$ to $N$ **do**
4:         $\mathrm{z}_j, \mathrm{y}_j \sim p_{\boldsymbol{\theta}}(\mathrm{z}, \mathrm{y}|\mathrm{x}^N)$                           ▷ sample example from model
5:         $\mathrm{x}_i^N \leftarrow (\mathrm{x}_i^N, \mathrm{z}_j, \mathrm{y}_j)$                                  ▷ update approximation context
6:     $\mathrm{x}_i \leftarrow ()$                                                                                 ▷ initialize replicant data
7:     **for** $j \leftarrow 1$ to $n$ **do**
8:         $\mathrm{z}_j, \mathrm{y}_j \sim p_{\boldsymbol{\theta}}(\mathrm{z}, \mathrm{y}|\mathrm{x}_i^N)$                         ▷ sample example from model
9:         $\mathrm{x}_i \leftarrow (\mathrm{x}_i, \mathrm{z}_j, \mathrm{y}_j)$                                       ▷ update replicant data
10: **return** $\frac{1}{\mathrm{M}} \sum_{i=1}^{\mathrm{M}} \mathbb{1}\{g_{\boldsymbol{\theta}}(\mathrm{x}_i, \mathrm{x}_i^N) \geq g_{\boldsymbol{\theta}}(\mathrm{x}^{\text{test}}, \mathrm{x}_i^N)\}$        ▷ estimate $p$-value

---

We derive an estimator for the generative predictive $p$-value in Equation (3) that uses Monte Carlo estimates to approximate the integrals. Algorithm 1 describes the estimation procedure. The key stage that differentiates the generative predictive $p$-value algorithm from the standard posterior predictive $p$-value algorithm is described in Lines 2 to 5. Here datasets $\mathrm{x}_i^N$ of length $N$ are ancestrally sampled from the CGM predictive distribution to approximate sampling a mechanism $\mathrm{f}_i$. This is in contrast to sampling an explanation directly from the model posterior as shown in Algorithm 2 Line 2 of Appendix C. When sampling replication data $\mathrm{x}_i$ in Lines 6 to 9, the CGM predictive distribution is conditioned on $\mathrm{x}_i^N$ and $n$ new samples are independently generated. This procedure is repeated $M$ times, and the $p$-value is empirically estimated as before.

## 7 Empirical evaluation

This section reports the following empirical findings: (1) The generative predictive $p$-value is an accurate predictor of model capability in tabular, natural language, and imaging ICL problems. (2) The $p$-value computed under the NLL discrepancy is also an indicator of whether there are enough in-context examples $n$. (3) The number of generated examples $N - n$ interpolates the $p$-value between the posterior predictive $p$-value under the NLML discrepancy and the NLL discrepancy using the model posterior $p_{\boldsymbol{\theta}}(\mathrm{f} \mid \mathrm{x}^n)$ and likelihood $p_{\boldsymbol{\theta}}(\mathrm{x} \mid \mathrm{f})$. These findings show that the $p$-value computed under either discrepancy yields an accurate predictor of whether generative AI can solve your in-context learning problem. If you also need to know whether there are enough in-context examples, we suggest using the NLL discrepancy function. If computational efficiency is a primary concern, we suggest using the NLML discrepancy as dataset completion generation is not required.

**Models.** We evaluate our methods using two model types. For tabular and imaging tasks, we use a Llama-2 regression model for sequences of continuous variables [35]. The model is optimized from scratch for next token (variable or pixel) prediction following the procedure of Touvron et al. [8]. For natural language tasks, we use pre-trained Llama-2 7B [8] and Gemma-2 9B [50] LLMs (Gemma-2 9B results are reported in Appendix F).

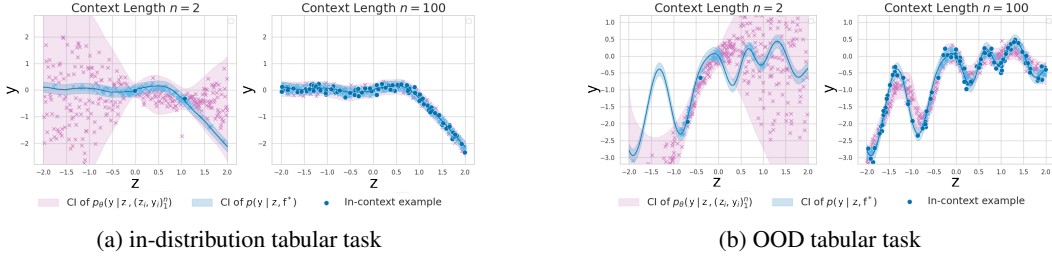

(a) in-distribution tabular task                                            (b) OOD tabular task

Figure 2: Tabular data tasks.

**Data.** For tabular tasks, queries z are sampled uniformly from the interval $[-2, 2]$. Responses y are drawn from a normal distribution with a mean $\mu(\mathrm{z})$, parameterized by either a random 3rd-degree

polynomial (in-distribution), a random ReLU neural network (in-distribution or OOD), or a radial basis function (RBF) kernel Gaussian process with a length scale of 0.3 (OOD). The training data comprise 8000 unique in-distribution datasets with 2000 $z - y$ examples each. An in-distribution ReLU-NN task is illustrated in Figure 2a. The mean function $\mu(z)$ is plotted by the blue line, and the blue shaded region outlines the 95% CI of $p_{\theta}(y \mid z, f^*)$. An OOD GP task is illustrated in Figure 2b. In-distribution test data comprise a set of 200 new random datasets with 500 $z - y$ examples each. The OOD test data comprise 200 random datasets with 500 $z - y$ examples each.

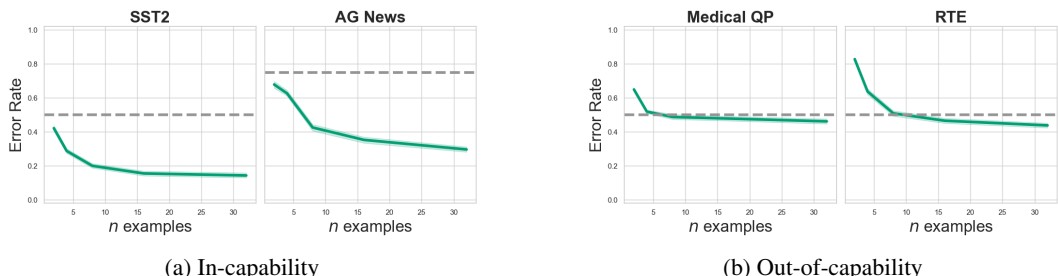

(a) In-capability

(b) Out-of-capability

Figure 3: Natural language in-capability vs. out-of-capability tasks. Green solid line is the ICL error rate for Llama-2 7B. Gray dashed line is the random guessing error rate.

For pre-trained LLM experiments, the delineation between in- and out-of-distribution is opaque. Instead, we use in-capability or out-of-capability to differentiate between tasks a model can or cannot perform well. Figure 3a illustrates in-capability tasks where the error rate of Llama-2 7B is considerably better than random guessing. The in-capability data are the SST2 [9] sentiment analysis (positive vs. negative) and AG News Zhang et al. [51] topic classification (World, Sports, Business, Sci/Tech) datasets. Figure 3b illustrates out-of-capability tasks where the error rate is only marginally better than random. The out-of-capability data are the Medical Questions Pairs (MQP) [10] differentiation (similar vs. different) and RTE [52] natural language inference (entailment vs. not entailment) datasets.

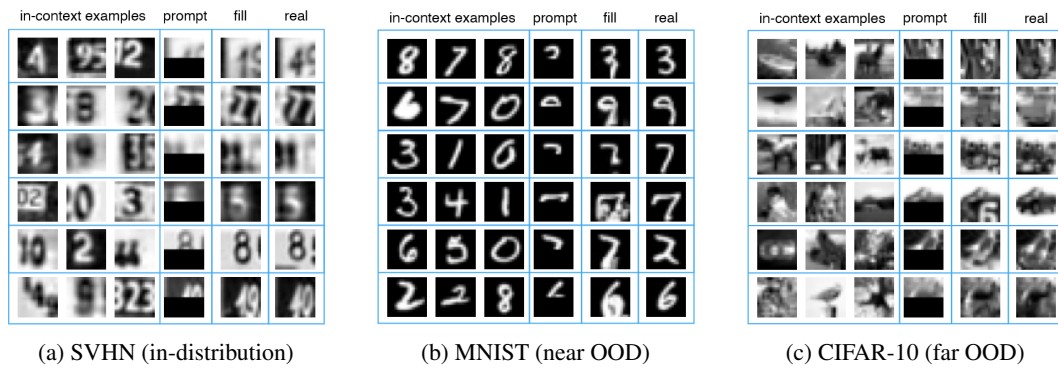

(a) SVHN (in-distribution)

(b) MNIST (near OOD)

(c) CIFAR-10 (far OOD)

Figure 4: Generative fill tasks using the test sets of SVHN, MNIST, and CIFAR-10.

For imaging ICL experiments, we use SVHN for in-distribution data [53], MNIST as "near" OOD data [54], and CIFAR-10 as "far" OOD data [55]. Our Llama-2 regression model takes a sequence of flattened, grayscale, 8x8 images as input. It is fit to random sequences of 16 images from the SVHN "extra" split, which has over 500k examples. A series of in-distribution generative fill tasks is shown in Figure 4a. In each row, the model is prompted with three in-context examples and asked to complete the missing half of the 4th example. Each completion in the "fill" column is sensible, even when the completed number differs from the "real" number. Figure 4b illustrates completions for near-OOD MNIST tasks. We see in rows 1, 2, 3, and 6 that the fills are often sensible, but the model is prone to hallucinating odd completions (row 4) and artifacts (row 5). Figure 4c illustrates completions for far-OOD CIFAR-10 tasks. The completions are surprisingly consistent at this resolution, but as the result in row 4 demonstrates, the model hallucinates completions from its domain.

**Discrepancy functions.** We evaluate the $p$-value using discrepancy functions defined as

$$g_{\boldsymbol{\theta}}(x, x^{(\cdot)}) := -\frac{1}{|x|} \sum_{z_i, y_i \in x} \frac{1}{|(z_i, y_i)|} \sum_{t_j \in (z_i, y_i)} \log p_{\boldsymbol{\theta}}(t_j \mid t_{<j}, x^{(\cdot)}),$$

where $|(z_i, y_i)|$ is the number of tokens in the evaluated example. Following this template, the per-token negative log marginal likelihood (NLML) is written $g_{\boldsymbol{\theta}}(x, x^n)$ and an estimate of the per-token the negative log-likelihood (NLL) is written $g_{\boldsymbol{\theta}}(x, x^N)$, where $x^N$ is generated as in Algorithm 1.

**Predicting model capability.** The $p$-values are calculated using either Algorithm 1 or Algorithm 3 and a significance level $\alpha$ is selected to yield a binary predictor of model capability $\mathbb{1}\{p_{\text{gpc}} < \alpha\}$; a model is predicted as incapable of solving the ICL problem if the estimated generative predictive p-value is less than the significance level. We report results for significance levels $\alpha \in [0.01, 0.05, 0.1, 0.2, 0.5]$. For the NLL discrepancy function, replication data x is independently sampled from the likelihood under a hypothetical dataset completion $p_{\boldsymbol{\theta}}(z, y \mid x^N)$. For the NLML discrepancy function, replication data is independently sampled from the predictive distribution $p_{\boldsymbol{\theta}}(z, y \mid x^n)$.

**Evaluation metrics.** We evaluate the capability predictor using standard metrics: FPR measures in-capability tasks misclassified as out-of-capability, Precision reflects correctly identified out-of-capability tasks, and Recall measures correctly detected out-of-capability tasks. F1 Score and Accuracy assess overall performance (see Figure 5 for definitions).

We also provide the distribution of $p$-values across tasks to assess how confidently the model distinguishes between the different

Figure 5: Evaluation metrics for GPC performance.

| Metric | Equation |
|---|---|
| False Positive Rate (FPR) | $\frac{\text{False Positives}}{\text{False Positives}+\text{True Negatives}}$ |
| Precision | $\frac{\text{True Positives}}{\text{True Positives}+\text{False Positives}}$ |
| Recall | $\frac{\text{True Positives}}{\text{True Positives}+\text{False Negatives}}$ |
| F1 Score | $\frac{2 \cdot \text{Precision} \cdot \text{Recall}}{\text{Precision}+\text{Recall}}$ |
| Accuracy | $\frac{\text{True Positives}+\text{True Negatives}}{\text{Total Number of Predictions}}$ |

ICL problems. Lower $p$-values indicate stronger confidence that a model cannot solve a problem.

## 7.1 The generative predictive $p$-value accurately predicts model capability

**Tabular data.** We first evaluate whether the generative predictive $p$-value effectively predicts OOD tabular data tasks. The parameters for Algorithm 1 are $M = 40$ replications and $N - n = 200$ generated examples. The ICL dataset $x^n$ size is varied from $n = 2$ to $n = 200$. Figure 6 plots precision, recall, F1, and accuracy curves and shows that the $p$-value estimates under either discrepancy function provide non-trivial OOD predictors for all $\alpha$ settings.

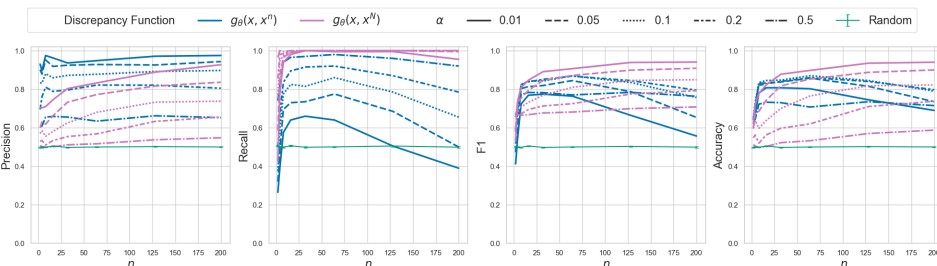

Figure 6: Tabular OOD detection. Metric values vs. context length. In-distribution functions are from unseen random ReLU-NNs. OOD functions are from an RBF kernel GP.

**Natural language ICL.** Next, we evaluate whether the generative predictive $p$-value effectively predicts out-of-capacity natural language tasks. The parameters for Algorithm 1 are $M = 20$ replications and $N - n = 10$ generated examples. The ICL dataset $x^n$ size is varied from $n = 4$ to $n = 64$. Figure 7 plots precision, recall, F1, and accuracy curves and shows that the $p$-value estimates under the NLL discrepancy provide non-trivial (accuracy $> 0.5$) out-of-capability predictors in the domain of natural language for all $\alpha$ settings. The NLML discrepancy $g_{\boldsymbol{\theta}}(x, x^n)$ is also generally robust outside of the small $n$ and small $\alpha$ setting.

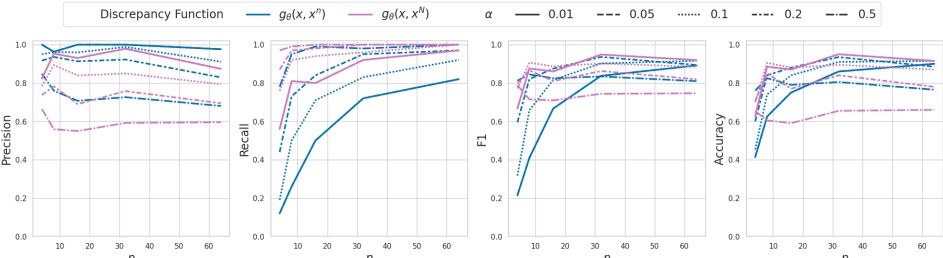

Figure 7: Llama-2-7B out-of-capability detection. Metric values vs. context length. In-capability tasks are from SST2 and AG News datasets. Out-of-capability tasks are from RTE and MQP datasets.

**Generative fill.** Finally, we evaluate whether the generative predictive $p$-value effectively predicts OOD generative fill tasks. The parameters for Algorithm 1 are $M = 100$ replications and $N - n = 8$ generated examples The ICL dataset $\mathbf{x}^n$ size is varied from $n = 2$ to $n = 8$. Figure 8 plots the OOD prediction metric curves and shows that the $p$-value estimates under either discrepancy function provide non-trivial (accuracy $> 0.66\bar{6}$) OOD predictors for all $\alpha$ settings.

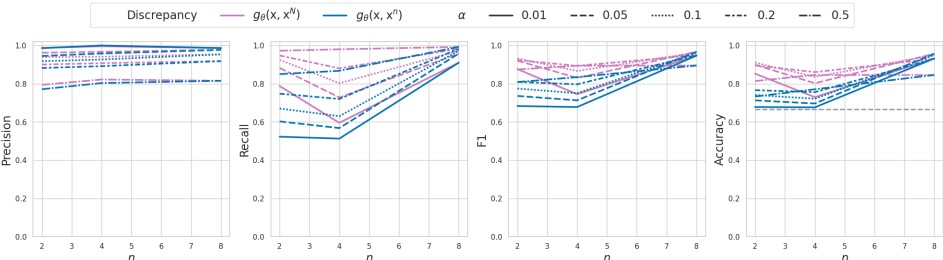

Figure 8: Generative fill OOD detection. Metric values vs. context length. In-distribution tasks are from the SVHN test set. Near and far OOD tasks are from the MNIST and CIFAR-10 test sets.

**Discussion.** Figures 6 to 8 reveal several trends. First, the NLML discrepancy (blue) yields better precision, indicating that it is less likely to misclassify an in-capability ICL problem as unsolvable. Second, the NLL discrepancy (purple) yields higher recall, indicating that it is less likely to misclassify an out-of-capability ICL problem as solvable. Third, the NLL discrepancy with significance level $\alpha = 0.05$ yields a generally more accurate predictor than the NLML discrepancy function for any significance level in the set evaluated. Finally, the recall of a predictor under the NLML discrepancy is sensitive to the number of in-context examples $n$. Next, we look deeper into the relationship between dataset size and the discrepancy functions.

## 7.2 The NLL discrepancy also indicates whether you have enough data

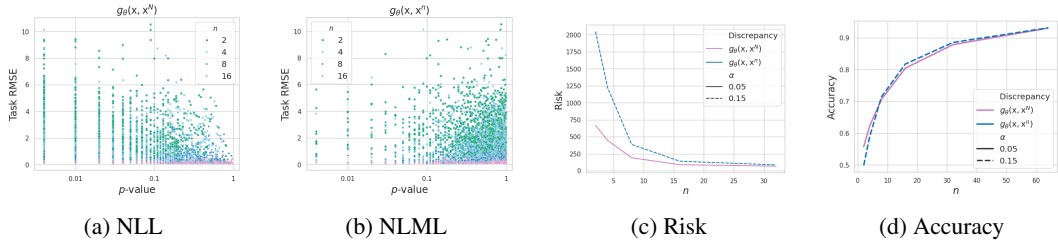

Figure 9: (a) and (b) Scatter plots of response RMSE vs. $p$-values for the NLL and NLML discrepancies. Points are style-coded by ICL dataset size $n$. (c) Risk vs. $n$. (d) Accuracy vs. $n$

Both discrepancy functions yield accurate predictors of model capability, but the NLL discrepancy also provides information about whether there are enough in-context examples to reliably solve a task. We use prediction RMSE over task responses to measure reliability. Figures 9a and 9b plot the

RMSE against the $p$-values computed under the NLL and NLML discrepancies for in-distribution polynomial tabular tasks. We see that lower $p$-values correlate with higher RMSE for the NLL discrepancy, but not for the NLML discrepancy. This added information is useful for reducing risk in recommendation systems that autonomously respond if the $p$-value is greater than the significance level $\alpha$. For example, at $\alpha = 0.1$, the NLL discrepancy reduces the generation of responses with higher error because it accounts for the number of examples provided. Taking the risk as the sum of task RMSEs for tasks predicted as in-capability, Figures 9c and 9d show that the NLL discrepancy results in substantially reduced risk, even when we closely match the accuracies of each predictor. Figure 13 in the appendix gives further insight into how the distributions of $p$-values evolve with dataset size for each discrepancy function.

### 7.3 The number of generated examples $N - n$ interpolates the $p - value$ estimate between the NLML and the ideal NLL discrepancies

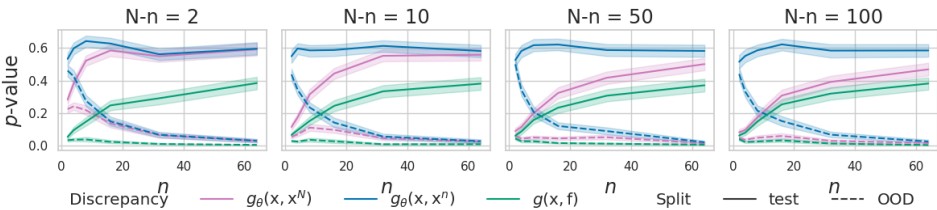

Figure 10: The dataset completion size $N - n$ interpolates the $p_{\mathrm{gpc}}$ under $g_{\boldsymbol{\theta}}(\mathrm{x}, \mathrm{x}^N)$ between the $p_{\mathrm{ppc}}$ under $g_{\boldsymbol{\theta}}(\mathrm{x}, \mathrm{x}^n)$ (NLML) and the $p_{\mathrm{ppc}}$ under $g_{\boldsymbol{\theta}}(\mathrm{x}, \mathrm{f})$ (NLL).

Inspection of Equations (1) to (3) makes clear that the dataset completion size $N - n$ should closely interpolate $p$-value estimates between $p_{\mathrm{ppc}}$ computed with the NLML discrepancy and with the NLL discrepancy using the likelihood and posterior of a Bayesian model. To verify this, we use a reference Bayesian polynomial regression model to compute the $p_{\mathrm{ppc}}$. We use our Llama-2 regression model fit to datasets generated from the reference model likelihood under different explanations to compute the $p_{\mathrm{gpc}}$. We let datasets generated by random ReLU-NNs serve as OOD tasks. Figure 10 demonstrates that our expectation is true. Specifically, the $p$-value estimates at $N - n = 2$ are distributionally close to those calculated under the NLML, and they more closely approximate those calculated under the reference NLL discrepancy as we increase $N - n$ to 100. The latter observation is also illustrated in Figure 12.

Since the $p$-values computed under either discrepancy yield accurate predictors of model capability, the choice between discrepancy functions ultimately comes down to a decision on whether the added computational cost of generating dataset completions is justified. If you need to know whether there are enough in-context examples to generate an accurate response—a necessity in risk-sensitive applications—then we recommend using the NLL discrepancy function. If computational efficiency or the cost of response deferral are primary concerns—practical user experience concerns—we suggest using the NLML discrepancy.

## 8   Conclusion

This work introduces the *generative predictive $p$-value*, a metric for determining whether a Conditional Generative Model can solve an In-Context Learning problem. It extends Bayesian model criticism techniques like PPCs to generative models by sampling dataset completions from the model's predictive distribution to approximate sampling latent explanations from a Bayesian model posterior. Empirical evaluations on tabular, natural language, and imaging tasks show that the generative predictive $p$-value can effectively identify the limits of model capability, distinguishing between in-capability and out-of-capability tasks for models like Llama-2 7B and Gemma-2 9B. This approach is a practical method to assess model suitability that advances Bayesian model criticism for CGMs. While we have focused on model capability prediction, the $p$-value estimates could also be used for model selection or as a general measure of task uncertainty. We are eager to explore extensions beyond ICL tasks to improve the reliability of generative AI systems.

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

# A Model intuition

In this section we give an interpretation of the component parts of a Bayesian model, how they are used to make inferences about uncertainty, and how to relate inferences in classical domains to inferences in more complex domains like language.

**The model prior** $p_{\boldsymbol{\theta}}(\mathrm{f})$ can be thought of as a library over the possible explanations a model could ascribe to observations. It is a special kind of library, where the probability of finding an explanation in the library at random is also defined. The model prior encodes everything "known" to a model; all the latent patterns available as explanations for—or an index of all the probability distributions ascribable to—any set of observations. The model prior may or may not assign non-zero probability to an explanation $\mathrm{f}$ equivalent to a given ICL problem task $\mathrm{f}^*$. If no such explanation has coverage under the prior, then the model may not be able to provide an accurate solution to the ICL problem.

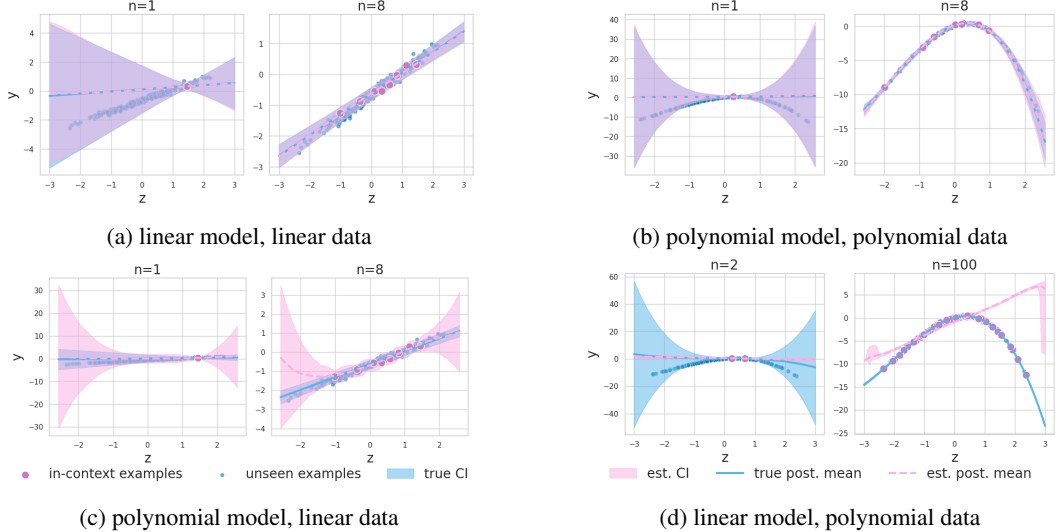

(a) linear model, linear data  (b) polynomial model, polynomial data

(c) polynomial model, linear data  (d) linear model, polynomial data

Figure 11: Examples of misaligned model and data combinations. Transformer models (pink) are fit to either linear or polynomial noisy data defined by reference Bayesian models (blue).

For example, a Bayesian linear regression model with fixed noise, defines the part of the prior over explanations $p_{\boldsymbol{\theta}}(\mathrm{f})$ related to the outcome $\mathrm{y}$ as a set of coefficient vectors; a set of hyperplanes. If the ICL dataset and prediction task are characterized by a linear relationship between continuous valued queries and responses—illustrated in Figure 11a—the prior would be suitable for the ICL problem. However, if they are characterized by polynomial relationships—illustrated in Figure 11c—then the relationship would not have coverage under the prior and the precision of responses under the model would be limited by the capacity of a hyperplane to fit a polynomial surface. By analogy, a LLM that is pre-trained or fine-tuned on a large set of integrals expressed in natural language may have the functional capacity to integrate; generalize to unseen functions and domains of the classes and spaces covered in the training set. So if the ICL dataset and task are related to the integration of polynomials, the learned library of mappings from text to token distributions may be appropriate for that ICL problem. However, if the LLM training corpus did not contain content related to calculus, then learned library of mappings may not include the functional capability to solve the ICL problem.

**The model likelihood** $p_{\boldsymbol{\theta}}(\mathrm{x} \mid \mathrm{f})$ encodes the variety of observations $\mathrm{x}$ that could be generated according to a given explanation $\mathrm{f}$. The variety encoded by this distribution is often called *aleatoric* uncertainty—aleatory is a pretentious word for random—which refers to the inherent *randomness* over generated datasets when sampling according to the likelihood under a given explanation $\mathrm{f}$. For example, given the explanation implied by a fair coin, we will still be *uncertain* whether the outcome of a single coin flip will be heads or tails. More contemporarily, if you are already familiar with this concept and I were to say, "I would like to share the idea of aleatoric uncertainty with you," you would know *which* idea I want to share, but, before reading this paragraph, you would be uncertain about *how* I would share it with you. When the model likelihood $p_{\boldsymbol{\theta}}(\mathrm{x} \mid \mathrm{f})$ is indexed by an explanation that is equivalent to the task $\mathrm{f} \equiv \mathrm{f}^*$, then $p_{\boldsymbol{\theta}}(\mathrm{x} \mid \mathrm{f})$ is equal to the reference likelihood $p(\mathrm{x} \mid \mathrm{f}^*)$. So

even though we may be uncertain about *which* dataset would be generated according to the model likelihood, we could still be certain that the generated dataset would correspond to the task. However, if there is a discrepancy between the model and reference likelihood then the model may not be suitable for the in-context learning problem.

**The model posterior**

$$p_{\boldsymbol{\theta}}(f \mid x^n) = \frac{p_{\boldsymbol{\theta}}(x^n \mid f)p_{\boldsymbol{\theta}}(f)}{\int p_{\boldsymbol{\theta}}(x^n \mid f)d\mathrm{P}_{\boldsymbol{\theta}}(f)},$$

derived from the model joint $p_{\boldsymbol{\theta}}(x, f)$ via Bayes's theorem, encodes variety over explanations that *could* have generated a specific set of observations $x^n$. This variety is often called *epistemic* uncertainty. Epistemic is a pretentious term referring to knowledge, conveying that we may not yet know *which* explanation $f$ among subset of reasonable explanations best explains possible datasets $x$ under the ICL problem.

For example, given only four observations—say, two heads and two tails—the sample mean estimate for the probability of observing heads is 0.5. However, we may still be uncertain about whether the coin generating the outcomes was fair or biased because the variance of that estimate is still a non-negligible 0.0625 when we assume the coin is actually fair. Related to our contemporary example, if I only say, "I would like to share an idea with you," you can probably imagine an abundance of ideas that I could be referring to and thus still be uncertain about which one I have chosen to share.

A relevant feature of epistemic uncertainty is that it is reducible as we observe more context. In the coin flip example, as we observe more outcomes, our certainty about the probability of heads increases. In the second example, you may have a better idea about the class of ideas I may share based on what has been presented thus far.

As a function of both the model likelihood and prior, the model posterior inherits the limitations of both. But it also provides information about whether an in-context learning problem can be solved reliably. Namely, variety over explanations is indicative of being uncertain about which task the ICL dataset corresponds to. This variety can lead to the model generating responses corresponding to tasks other than $f^*$. But it may also be indicative of when more examples (larger $n$) can improve the quality of solutions to an in-context learning problem.

**The model posterior predictive**

$$p_{\boldsymbol{\theta}}(x \mid x^n) = \int p_{\boldsymbol{\theta}}(x \mid f)d\mathrm{P}_{\boldsymbol{\theta}}(f \mid x^n),$$

is derived from the model to generate new observations $x$ given past observations $x^n$. Poetically, the model posterior predictive gives the model a voice to respond to observations with observations. The model posterior predictive convolves the model likelihood of the observations given an explanation with the model posterior over explanations. This process entangles variety over explanations after observing a dataset $x^n$ and variety over observations $x$ for each specific explanation $f$; the model posterior predictive entangles aleatoric and epistemic uncertainty.

**Model inferences.** A model $\boldsymbol{\theta}$ defined over an observation space $\mathcal{X}$ is used make inferences about observations from that space $x^n$. Inferences like the probability of the the next word given a sequence of words, model uncertainty, and countless other things. But a model is a choice—the practitioner makes a modeling decision—and so the inferences derived from observations under a model may or may not be grounded in reality.

Figure 11 illustrates inferences about the predictive distribution made by different models given different datasets. When the data and model are well aligned— Figures 11a and 11b—model inferences (pink) overlap with those made by the reference model (blue) and appear purple. However, inferences made by a misaligned polynomial model—Figure 11c—are much wider that those made by the reference linear model. And inferences made by a misaligned linear model—Figure 11d—are more narrow than those ogf the reference polynomial model, which results in the model being confident and wrong. Similarly, a very general LLM may have compromised data efficiency for rare domains, while a highly specialized LLM may no longer generalize beyond its domain.

These examples illustrate that while models are used to quantify empirical facts—like the frequency of an event occurring—they also carry a subjective aspect that needs to be considered when using model inferences in practice. This consideration guides our question of when a model will provide reliable inferences for a given ICL problem.

# B  Proofs for theoretical results

We restate our formalization and assumptions for convenience. Observable examples $(z, y) \in \mathcal{X}$ are modeled by the $(\mathcal{X}, \mathcal{A})$-random variable $X_i$ and explanations $f \in \mathcal{F}$ are modeled by the $(\mathcal{F}, \mathcal{B})$-random variable $F$, where $\mathcal{A}$ and $\mathcal{B}$ are the relevant sigma algebras. For each $f \in \mathcal{F}$, let the model $\boldsymbol{\theta}$ define a probability measure $P_{\boldsymbol{\theta}}^f$ on $(\mathcal{X}, \mathcal{A})$. Let the model $\boldsymbol{\theta}$ further define a probability measure $P_{\boldsymbol{\theta}}$ on $(\mathcal{F}, \mathcal{B})$. And let $P_{\boldsymbol{\theta}}$ and $P_{\boldsymbol{\theta}}^f$ define the joint measure $M_{\boldsymbol{\theta}}$ over $((X_1, X_2, \dots), F)$. Our method rests on Doob's theorem for estimators [56], which assumes the following three conditions.

**Condition 1.** *The observation $\mathcal{X}$ and explanation $\mathcal{F}$ spaces are complete and separable metric spaces.*

**Condition 2.** *The set of probability measures $\{P_{\boldsymbol{\theta}}^f : f \in \mathcal{F}\}$ defined by the model $\theta$ is a measurable family; the mapping $f \mapsto P_{\boldsymbol{\theta}}^f(A)$ is measurable for every $A \in \mathcal{A}$.*

**Condition 3.** *The model $\boldsymbol{\theta}$ is identifiable;*

$$f \neq f' \Rightarrow P_{\boldsymbol{\theta}}^f \neq P_{\boldsymbol{\theta}}^{f'}. \tag{4}$$

We state Doob's theorem for convenience.

**Theorem 2. *Doob's Theorem for estimators.*** *Let $F \sim P_{\boldsymbol{\theta}}$ and $X_1, X_2, \dots$ i.i.d $\sim P_{\boldsymbol{\theta}}^f$. Then under general conditions on identifiability, $\mathcal{F}$ and $\mathcal{X}$ (see Appendix B), and a measurable function $h : \mathcal{F} \to \mathbb{R}$ such that $\int |h(f)| dP_{\boldsymbol{\theta}}(f) < \infty$, then*

$$\lim_{n \to \infty} \mathbb{E}[h(F) \mid X_1, X_2, \dots, X_n] = h(F) \; a.s. \; [M_{\boldsymbol{\theta}}]. \tag{5}$$

*Proof.* Miller [57] provides a detailed proof of this theorem. $\qquad\square$

**Lemma 3.** *Let $F \sim P_{\boldsymbol{\theta}}$, and $X_1, X_2, \dots$ i.i.d $\sim P_{\boldsymbol{\theta}}^f$. Assume Conditions 1 to 3 and let $\{\int |\log p_{\boldsymbol{\theta}}(x^m \mid f)| dP_{\boldsymbol{\theta}}(f) < \infty : \forall x^m \in \mathcal{X}^m\}$. Then,*

$$g_{\boldsymbol{\theta}}(x, F) = -\frac{1}{|x|} \log p_{\boldsymbol{\theta}}(x \mid F) = -\frac{1}{|x|} \log p_{\boldsymbol{\theta}}(x \mid X^\infty) = g_{\boldsymbol{\theta}}(x, X^\infty).$$

*Proof.*

$$
\begin{aligned}
p_{\boldsymbol{\theta}}(x \mid F) &= \lim_{n \to \infty} \int p_{\boldsymbol{\theta}}(x \mid f) dP_{\boldsymbol{\theta}}(f \mid X_1, \dots, X_n) \\
&= \lim_{n \to \infty} \int p_{\boldsymbol{\theta}}(x \mid f, X_1, \dots, X_n) dP_{\boldsymbol{\theta}}(f \mid X_1, \dots, X_n) \\
&= \lim_{n \to \infty} p_{\boldsymbol{\theta}}(x \mid X_1, \dots, X_n)
\end{aligned}
$$

$$
\begin{aligned}
g(x, F) &= -\frac{1}{|x|} \log p_{\boldsymbol{\theta}}(x \mid F) \\
&= -\frac{1}{|x|} \log \lim_{n \to \infty} p_{\boldsymbol{\theta}}(x \mid X_1, \dots, X_n) \\
&= \lim_{n \to \infty} -\frac{1}{|x|} \log p_{\boldsymbol{\theta}}(x \mid X_1, \dots, X_n) \\
&= -\frac{1}{|x|} \log p_{\boldsymbol{\theta}}(x \mid X^\infty) \\
&= g(x, X^\infty)
\end{aligned}
$$

$\qquad\square$

**Theorem 2.** *Under the conditions of Lemma 3,*

$$p_{ppc} = p_{mpc}$$

*Proof.* Define an alternative probability model such that $F \sim P_{\boldsymbol{\theta}}^{x^n}$ and $X_1, X_2, \ldots$ i.i.d $\sim P_{\boldsymbol{\theta}}^{f,x^n}$. Let $p_a$, $P_b$, and $g_a$ denote the relevant quantities respecting this model. For example, $p_a(y \mid x) = p_{\boldsymbol{\theta}}(y \mid x, x^n)$ and $P_a(f) = P_{\boldsymbol{\theta}}(f \mid x^n)$. Note that $p_{\boldsymbol{\theta}}(x \mid f) = p_{\boldsymbol{\theta}}(x \mid f, x^n) = p_a(x \mid f)$ since $X$ and $X^n$ are independent when f is known.

$$
\begin{aligned}
p_{\text{ppc}} &= \iint \mathbb{1}\left\{g_{\boldsymbol{\theta}}(x, f) \geq g_{\boldsymbol{\theta}}(x^{\text{test}}, f)\right\} d\mathrm{P}_{\boldsymbol{\theta}}(x \mid f) d\mathrm{P}_{\boldsymbol{\theta}}(f \mid x^n) \\
&= \int \mathbb{1}\left\{g_a(x, f) \geq g_a(x^{\text{test}}, f)\right\} dP_a(x, f) \\
&= \int \mathbb{1}\left\{g_a(x, f) \geq g_a(x^{\text{test}}, f)\right\} dP_a(x, f, x^{n+1:\infty}) \\
&= \iint \mathbb{1}\left\{g_a(x, f) \geq g_a(x^{\text{test}}, f)\right\} dP_a(x, \mid f, x^{n+1:\infty}) dP_a(f, x^{n+1:\infty}) \\
&= \iint \mathbb{1}\left\{g_a(x, x^{n+1:\infty}) \geq g_a(x^{\text{test}}, x^{n+1:\infty})\right\} dP_a(x \mid f, x^{n+1:\infty}) dP_a(f, x^{n+1:\infty}) \quad \text{Lemma 3} \\
&= \iint \mathbb{1}\left\{g_a(x, x^{n+1:\infty}) \geq g_a(x^{\text{test}}, x^{n+1:\infty})\right\} dP_a(x \mid x^{n+1:\infty}) dP_a(x^{n+1:\infty}) \\
&= \iint \mathbb{1}\left\{g_{\boldsymbol{\theta}}(x, x^\infty) \geq g_{\boldsymbol{\theta}}(x^{\text{test}}, x^\infty)\right\} d\mathrm{P}_{\boldsymbol{\theta}}(x \mid x^\infty) d\mathrm{P}_{\boldsymbol{\theta}}(x^{n+1:\infty} \mid x^n) \\
&= p_{\text{mpc}}
\end{aligned}
$$

$\square$

## C  Posterior predictive p-value algorithm

Algorithm 2 details the procedure for calculating the posterior predictive p-value in Equation (1) given train data $x^n$, test data $x^{\text{test}}$, a discrepancy function $g_{\boldsymbol{\theta}}(x, f)$, and the nuber of replication datasets to generate $M$.

---

**Algorithm 2** $\widehat{p}_{\text{ppc}}$

---

**Require:** data $\{x^n, x^{\text{test}}\}$, discrepancy function $g_{\boldsymbol{\theta}}(x, f)$, # replicates M
1: **for** $i \leftarrow 1$ to M **do**
2:     $f_i \sim p_{\boldsymbol{\theta}}(f \mid x^n)$                                         ▷ sample explanation f
3:     $x_i \leftarrow ()$                                                      ▷ initialize replicant data
4:     **for** $j \leftarrow 1$ to $n$ **do**
5:         $z_j, y_j \sim p_{\boldsymbol{\theta}}(z, y \mid f_i)$                        ▷ sample example from model likelihood
6:         $x_i \leftarrow (x_i, z_j, y_j)$                                  ▷ update replicant data
7: **return** $\frac{1}{M} \sum_{i=1}^{M} \mathbb{1}\left\{g_{\boldsymbol{\theta}}(x_i, f_i) \geq g_{\boldsymbol{\theta}}(x^{\text{test}}, f_i)\right\}$                ▷ estimate p-value

---

## D  Lite generative predictive p-value algorithm

---

**Algorithm 3** $\widehat{p}_{\text{gpc}}^{\text{lite}}$

---

**Require:** data $\{x^n, x^{\text{test}}\}$, a discrepancy function $g_{\boldsymbol{\theta}}(x, x^n)$, # replicates M
1: **for** $i \leftarrow 1$ to M **do**
2:     $x_i \leftarrow ()$                                                      ▷ initialize replicant data
3:     **for** $j \leftarrow 1$ to $n$ **do**
4:         $z_j, y_j \sim p_{\boldsymbol{\theta}}(z, y \mid x_i, x^n)$                     ▷ sample example from model
5:         $x_i \leftarrow (x_i, z_j, y_j)$                                  ▷ update replicant data
6: **return** $\frac{1}{M} \sum_{i=1}^{M} \mathbb{1}\left\{g_{\boldsymbol{\theta}}(x_i, x^n) \geq g_{\boldsymbol{\theta}}(x^{\text{test}}, x^n)\right\}$                ▷ estimate p-value

---

Algorithm 3 summarizes a "lite" version of the estimator that forgoes approximate sampling from the model posterior and likelihood. Instead, it samples replication data directly from the model predictive

distribution and calculates the discrepancy functions with respect to the observed data $\mathrm{x}^n$ rather than a dataset completion $\mathrm{x}^N$.

# E  Additional figures

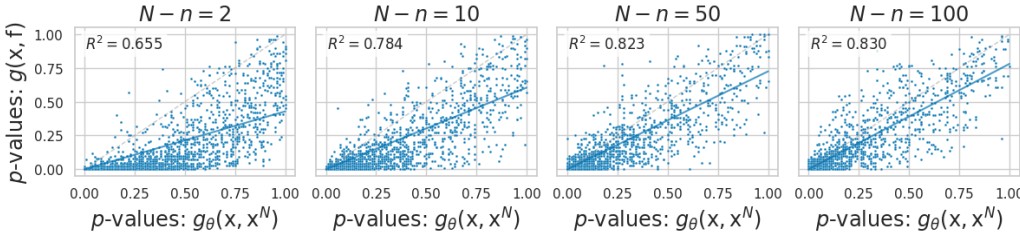

Figure 12: Scatter plots demonstrating that $p_{\mathrm{gpc}}$ becomes a better approximation of $p_{\mathrm{ppc}}$ with increasing dataset completion size $N - n$.

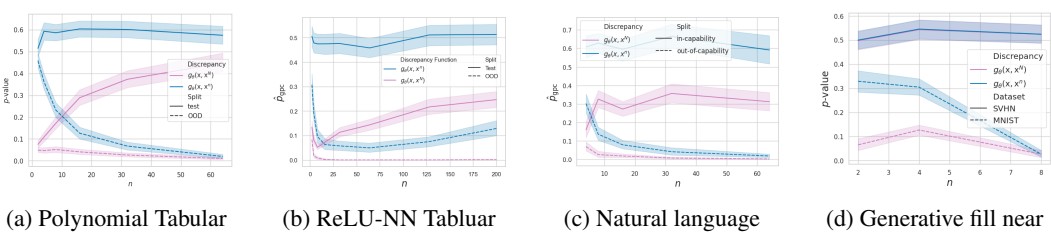

(a) Polynomial Tabular    (b) ReLU-NN Tabluar    (c) Natural language    (d) Generative fill near

Figure 13: The generative predictive $p$-value against dataset size $n$

# F  Gemma-2 9B results

Figure 14a shows that the in-capability vs. out-of-capability distinction is also sensible for the Gemma-2 9B model. So we conduct the same analysis for Gemma-2 9B that we did for Llama-2 7B in Section 7.

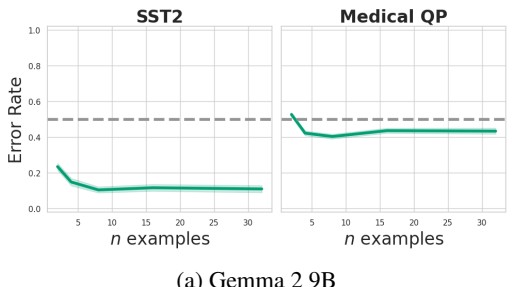

(a) Gemma 2 9B

Figure 14: Natural language in-capability vs. out-of-capability tasks.

Figure 15a shows $p$-values under the Gemma-2 9B model as a function of the ICL dataset $\mathrm{x}^n$ size $n$ (context length). We see a clear separation between the estimated generative predictive $p$-values $\widehat{p}_{\mathrm{gpc}}$ for the in-capacity SST2 data (solid lines) and the out-of-capacity MQP dataset (dashed lines), but only for the NLML discrepency. The separation is robust across different ICL dataset sizes.

Figure 15b plots the FPR for the capability detector defined by $\widehat{p}_{\mathrm{gpc}}$ the with NLML discrepancy. We do not see the same stability of the FPR across ICL dataset size $n$ that we saw for the Llama-2 7B model. Instead the FPR decreases with increasing $n$ for all significance level $\alpha$. Figure 15c plots the FPR for the capability detector defined by $\widehat{p}_{\mathrm{gpc}}$ the with NLL discrepancy. We see that the false positive rate is high for all values. These findings are reflected in the Precision curves on the left

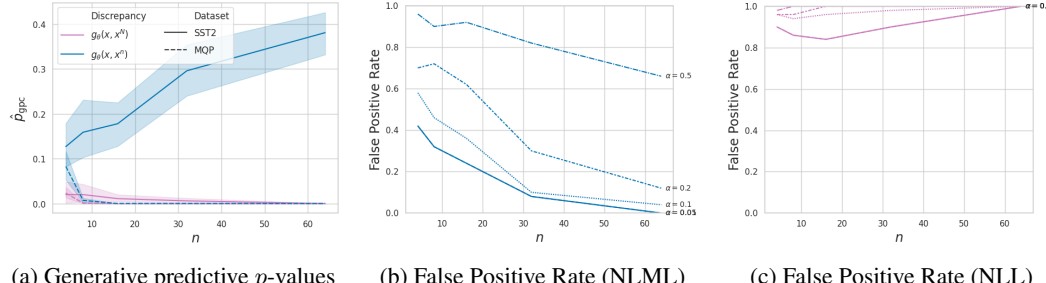

(a) Generative predictive $p$-values     (b) False Positive Rate (NLML)     (c) False Positive Rate (NLL)

Figure 15: Natural language task with Gemma-2 9B. The generative predictive $p$-value against dataset size $n$ and it's relationship to the false positive rate. The first figure shows the generative predictive $p$-values, and the second and third figures show the false positive rate with the NLML and NLL discrepancy functions, respectively.

hand side of Figure 16. We again see in the Recall curve that the NLL discrepancy leads to a more sensitive predictor than the NLML discrepancy. The F1 and Accuracy curves show that the NLML based $p$-value leads to a much more effective predictor for Gemma-2 9B.

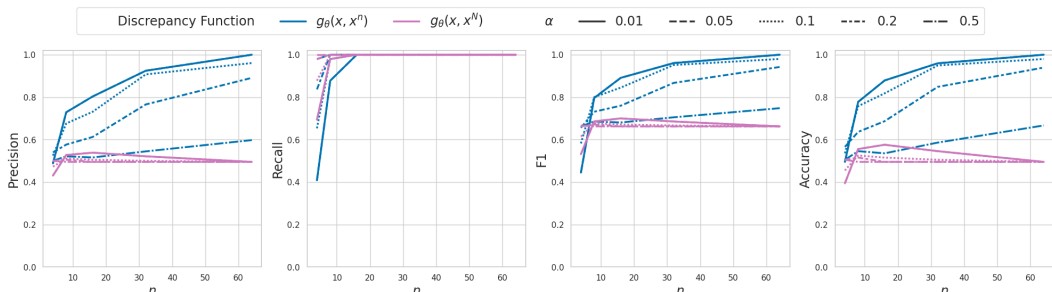

Figure 16: Natural language model suitability detection ablation. Precision, recall, F1, and accuracy metrics vs. number of in-context examples. SST2 ICL datasets are taken to be in-capability for Gemma-2 9b. MQP ICL datasets are taken to be out-of-capability for Gemma-2 9b.

