# OpenReview forum: "Can Generative AI Solve Your In-Context Learning Problem? A Martingale Perspective"
_NeurIPS.cc/2024/Workshop/SafeGenAi — SafeGenAi Poster_

### Official Review · Reviewer_QpLC · 2024-10-09
**This paper investigates whether a conditional generative model (CGM) like LLMs solve in-context learning (ICL) tasks. While traditional bayesian model's require access to the model's posterior distributions (which are unavailable for LLMs), this paper introduces the "generative predictive p-value". This is a novel measure that looks at the predictive distribution of the CGM.**

**Rating:** 8
**Confidence:** 3

**Review:**

Strengths:
This is a novel paper. With LLMs, we only have access to their predictive distributions, so a metric like the proposed generative predictive p-value would be super useful. The authors of the paper prove that the martingale predictive p-value is equivalent to the posterior predictive p-value using theoretical proofs. Furthermore, the authors also provide an algorithm for estimating the generative predictive p-value. Their method only requires the CGMs to generate queries and responses from. The authors also provide empirical validations by experimenting with regression and ICL tasks using llama-2 and gemma-2 models and reporting their performance on standard evaluation metrics. They show that their proposed method effectively predicts a model's ability to solve ICL. The paper has solid theoretical foundations (theoretical proofs that show that the martingale predictive p-value is equivalent to the posterior predictive p-value under certain conditions), which is one of its key strengths.

Areas to improve on:
1. The authors should consider including other models and ICL tasks in order to improve the generalizability of these findings.
2. I also wonder how this proposed method compares to other methods that are trying to tackle the same problem such as Ling's paper on Uncertainty Quantification for In-Context Learning of Large Language Models [1].
3. Please also include a more formal analysis of the estimation error due to finite N (that the authors briefly acknowledge), if possible.


References:
1. Ling, C., Zhao, X., Zhang, X., Cheng, W., Liu, Y., Sun, Y., ... & Chen, H. (2024, June). Uncertainty Quantification for In-Context Learning of Large Language Models. In Proceedings of the 2024 Conference of the North American Chapter of the Association for Computational Linguistics: Human Language Technologies (Volume 1: Long Papers) (pp. 3357-3370).

---

### Official Review · Reviewer_q4nm · 2024-10-09
**Very important proof and problem, Impressive**

**Rating:** 10
**Confidence:** 2

**Review:**

### Summary:
The paper proposes a method to assess whether a Conditional Generative Model (CGM) can solve an in-context learning (ICL) problem. It introduces the generative predictive p-value, extending Bayesian model criticism techniques like posterior predictive checks (PPCs) to contemporary generative models using ancestral sampling. The authors demonstrate the method's effectiveness on synthetic regression and natural language tasks, highlighting its utility in determining a model's suitability for specific ICL problems.
### Strengths
1. **Strict Theoretical Analysis:** The paper rigorously proves that ICL is not strictly Bayesian but can be approximated using Conditional Generative Models (CGMs). This clarification addresses a significant gap in understanding whether ICL aligns with Bayesian inference principles.
2. **Novel Approximation Approach:** The authors introduce a new metric, the generative predictive p-value, which enables the use of Bayesian model criticism techniques for CGMs even in the absence of explicit Bayesian components like priors and posteriors.
3. **Clear Theoretical Basis:** A very rigorous theoretical proof that ICL is not strictly Bayesian. (I'm not a professional)
4. **Practical Implementation:** The method's reliance on generating queries and responses from the CGM, followed by log probability evaluation, ensures that it remains feasible for real-world applications without needing complex adjustments to the CGM.
5. **Scalable and Robust:** The approach is applicable to large language models and has shown robustness across tasks with varying dataset sizes, indicating its adaptability to different contexts.

### Weaknesses
1. **Estimation Error:** As the article says, the analysis of Estimation Error is not complete. I look forward to seeing it in my future work.
2. **discrepancy function** As I understand it, the p-value is very dependent on the discrepancy function(NLML or NLL), which may cause problems.
3. **Complexity of Implementation:** Despite its practical aspects, the method might still be challenging for practitioners without a strong background in Bayesian statistics or machine learning, due to the advanced theoretical concepts like martingales involved.
### Recommendation
My research direction is not this major, so I am not sure about the correctness of the proof. But for the rest of the discussion, the paper makes a valuable contribution to the field of generative AI and in-context learning, offering a novel and theoretically sound method that bridges the gap between ICL and Bayesian inference. By rigorously proving that ICL is not strictly Bayesian but can be approximated using CGMs, the paper provides a clear pathway for applying Bayesian model criticism techniques to contemporary AI models. I therefore **strongly recommend acceptance**.

---

### Official Review · Reviewer_K9KV · 2024-10-09

**Rating:** 8
**Confidence:** 3

**Review:**

Summary:
This paper introduces a novel approach to assessing the reliability of Conditional Generative Models for In-Context Learning tasks. It applies and extends the concept of Bayesian Model criticism techniques like Posterior Probability Checks to Gen AI tasks by positing that when the ancestral sampling from the predictive distribution of a CGM is equivalent to sampling datasets from the posterior predictive of the assumed Bayesian model, we can compute a generative predictive p-value that can determine if a model is reliable from an ICL perspective.

Strengths:
1. The paper is well written and lucid and undertakes a deeply scientific approach for assessing the reliability of generative models.
2. The statistical arguments presented here are well researched and sound. The paper pushes the boundaries when it comes to the applicability of Bayesian Model criticism techniques to general use cases like GenAI. These efforts seem to show positive results for their use case.

Minor concerns:
1. While the work is promising and sound, I would like the editor to understand if the reliability assessment of a Gen AI falls under the workshop's definition of SafeGenAI.
2. The abstract and body of the paper could do with some simplification of sentences to make it more reader-friendly.

---

### Official Review · Reviewer_9G44 · 2024-10-10
**Novel idea to view ICL from a martingale perspective but still room for improvement**

**Rating:** 6
**Confidence:** 2

**Review:**

Summary: the paper takes a Bayesian perspective towards in-context learning (ICL) problem. The paper establishes the theorem that justify the use of ancestral sampling from the conditional generative model to address the unavailability of the posterior defined by the Bayesian model, which allows the authors to propose the generative predictive p-value, as a measure for when the model can solve an ICL problem. They demonstrate the effectiveness of their method on synthetic regression task as well as natural language tasks.

Strength: The paper provides and proves theorem that justify their proposed algorithm and the proposed generative predictive p-value.  They also conduct experiments to demonstrate the effectiveness of their proposed metric.

Weakness: 1. The paper does not include a related work section in the main text. I suggest adding the related works section to help the readers understand the relevant backgrounds. 2. The discussion on the theorem is a little too concise. It’s hard for me to get the connection to the martingale theory. I suggest adding some high-level discussions on the interpretation from a martingale perspective. 3. The figure captions are too concise. It is hard to understand the figure with only reading the captions. 4. For experiments with natural language tasks, I think there might be some circulation in logic. It seems unclear how the authors define in-capacity and out-of-capacity.

---

### Official Review · Reviewer_MmZo · 2024-10-11
**Review : Can Generative AI Solve Your In-Context Learning Problem? A Martingale Perspective**

**Rating:** 4
**Confidence:** 2

**Review:**

The paper introduces a new metric to assess how well a conditional generative model performs in solving an in-context learning problem.

While the task and methodology are interesting, it is difficult to fully evaluate the contribution due to the paper’s lack of clarity and readability.

Specifically:
- The paper is challenging to follow. The abstract is already full of acronyms, making the addressed problem and core contribution not completely clear.
- The structure could be improved. Despite having 8 sections, there is no dedicated related work section, which makes it harder to place the work in context. In general, the paper can be better organized.
- Figure 1 could benefit from a more detailed caption (e.g., clarifying that SST2 refers to sentiment analysis), even though this is explained later in the text.
- The connection between evaluating in-context learning and safety is somewhat implicit. I recommend discussing this relationship more explicitly to enhance the clarity of the argument.